# Designing Policy Mixes to Address the World's Worst Devastation of a Rural Landscape Caused by *Xylella* Epidemic

Antonio Lopolito *  and Edgardo Sica 

Department of Economics, Management and Territory (DEMeT), University of Foggia, Via Da Zara 11, 71121 Foggia, Italy; edgardo.sica@unifg.it
* Correspondence: antonio.lopolito@unifg.it

**Abstract:** The socio-economic consequences of the *Xylella fastidiosa* epidemic represent a global problem that can only be addressed through tailored, local solutions. The selection of public interventions is not a trivial task for policy makers, as they must weigh many different interests (e.g., private profit, ecosystem services, usability, preservation and growth of real estate value, amenities, and land protection). The present paper addresses this challenge by building participatory scenarios based on "fuzzy cognitive maps," with the aim of identifying effective, acceptable, and efficient policy mixes to address the *Xylella* epidemic. The work investigates the case of southern Salento (Italy)–an olive production area at the epicentre of the global *Xylella* outbreak–to identify the most suitable actions for regenerating the landscape. To this end, the most efficient policy mixes are determined according to three possible policy perspectives, which provide different weights for effectiveness and acceptability. The results show that the proposed methodological approach may assist policy makers in coping with multifaceted policy challenges.

**Keywords:** policy mixes; iso-utility sets; policy perspectives; *Xylella fastidiosa*; fuzzy cognitive maps

## 1. Introduction

*Xylella fastidiosa* (*Xf*) is an insect-vectored bacterial plant pathogen that originated on coffee plants in Costa Rica. Subsequently, it appeared in southern Italy, southern France, Spain (i.e., Balearic Islands), Germany, China, and Iran. Outbreaks cause a significant economic impact in the agricultural sector. The microorganism affects more than 350 species, including many woody and herbaceous plants, crops, and weeds [1]. Although many plants are asymptomatic hosts, in some species, *Xf* induces severe and even lethal changes, such as plume disease (in peach trees), oleander leaf scorch, and citrus canker. This characteristic, coupled with its capacity to spread through a large number of host plants, represents a tremendous threat to the agricultural landscape, given the widespread presence of susceptible species.

The threat of *Xf* has become most evident in Salento—a large area in the Apulia region (southern Italy), known for its centuries-old olive groves. Here, *Xf* has provoked rapid olive tree desiccation, resulting in one of the worst phytosanitary emergencies in the world. Approximately 22 million plants have been infected and nearly 6.5 million olive trees have died [2], generating significant (direct and indirect) economic and social consequences. Direct effects include the reduction of agricultural incomes, the contraction of agri-food production, and negative externalities throughout the value chain. Indirect effects stem from the severe destruction of the rural landscape (Figure 1), which has an adverse impact on tourism, property values, and quality of life for local residents. Indeed, beyond the direct use values generated, the olive groves in Salento provide recreational, aesthetic, and educational benefits that contribute to health, security, and social relations, thus improving residents' well-being [3]. These effects are the most difficult to analyze, due to the intricate interactions between various landscape elements.

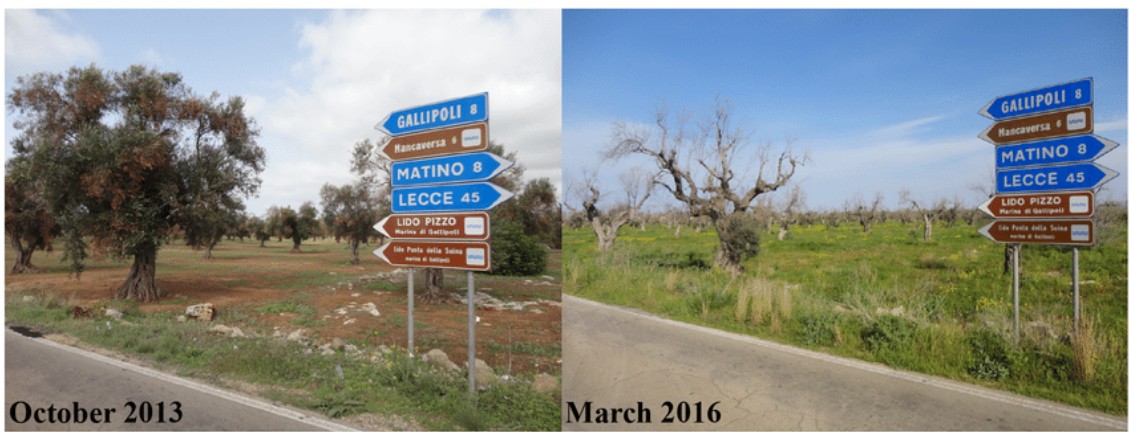

**Figure 1.** Salento's landscape devastation due to *Xylella fastidiosa* infection [4].

A landscape is a complex system in which a number of actors (e.g., farmers, producers, citizens, tourists, public institutions) interact with a variety of objectives (e.g., profit, ecosystem services, usability, the preservation and growth of real estate values, amenities, land protection); through this interaction, they generate particular system properties (relating to land use), such as agricultural, rural, urban, industrial, and touristic vocation, or a mixture of these [5]. Accordingly, an analysis of individual behaviours is not sufficient to understand the outcomes of a landscape system, as the system is, in part, determined by the interactions among actors [5].

In most planning practices, complexity is addressed through the identification of subsystems (i.e., ecological, landscape, and socio-economic), which are assigned specific objectives and actions [6]; however, this reductionist approach can lead to inconsistencies or conflict, stemming from the complex interactions among subsystems. For example, a policy intervention may have non-linear outcomes, due to the adaptive responses (i.e., strategic behaviour, ignorance, non-rationality) of actors to the policy and other context variables [7,8]. The *Xf* epidemic has disrupted local systems at the landscape, naturalistic-environmental, and socio-economic levels, dramatically changing both land use opportunities and living conditions. For this reason, it is essential that policy makers obtain comprehensive knowledge about the current state of these systems and the possible effects of any policy decisions on the local community and landscape.

The complex systems approach emphasises the importance of empirically capturing relevant knowledge from experiential actors and stakeholders [9]. The framework assumes that individuals who live and operate within and near the system have embedded knowledge of the system's components and mechanisms. Thus, local stakeholders are solicited to provide insight into the system state and functioning [10], in order to support policymakers in successfully designing preferred interventions.

Within the complex systems approach, the analytical tool of "fuzzy cognitive maps" (FCMs) may be applied to capture the complexity of a system. Compared with other techniques of soliciting embedded knowledge (e.g., Q methodology, [11]; multi-criteria decision analysis, [12]; full interviews; [10]), FCMs are more manageable and flexible. Thus, they are able to depict dynamic systems, as they acknowledge feedback and loops among variables and provide a representation and analysis of causal relationships [7]. Moreover, FCMs allow for a complete investigation of scenarios, even when only verbal information is available (and including when this information is expressed in non-technical and/or vague or imprecise language). This helps to overcome the typical challenge of a lack of data on emergent or novel phenomena. Finally, FCMs are a powerful simulation tool that can identify likely system evolution paths according to both natural trends and external interventions [13]. From a policy perspective, it is worth noting that external interventions are based on a rational framework and a shared stakeholder vision of the

system. This allows for relatively legitimated policies to be implemented, leading to improved sustainability within the policy cycle.

For these reasons, in the present study, we built participatory scenarios based on the FCMs approach, with the aim of identifying effective, acceptable, and efficient policy mixes for renewing the rural landscape of Salento, which has been significantly affected by the *Xf* epidemic. To this end, we primarily investigated the southern part of Salento, which represents the olive production area that has been most severely hit by *Xf*. We obtained a thorough understanding of local stakeholders' perceptions of the territory, identifying the most important system variables (including the interactions between these variables) and trends in the system evolution. Subsequently, we developed a fully shared scenario among stakeholders, thereby improving the selection of efficient protection/regeneration policy mixes among the most effective and/or acceptable ones and reducing the risk of policy failure.

The remainder of the paper is structured as follows: Section 2 describes the materials and methods; Section 3 presents and discusses the results; and, Section 4 ends with some concluding remarks.

## 2. Materials and Methods

The FCMs technique, which gathers and analyzes relevant information from local stakeholders on system variables and their interactions, was applied to create participatory scenarios. Furthermore, FCMs were used to analyze the future evolution of the system through the application of fuzzy inference, representing the collected information in a computational model [14]. First employed in psychology research [15], this participatory method was later extended to decision making problems using neural network inference [16]. The benefit of the technique is that, as stakeholders share critical information, decision makers gain a deeper understanding of local community needs; this allows them to develop effective, acceptable, and efficient policy interventions, while simultaneously reducing conflict [7].

FCMs frame actors' perceptions of a system in cognitive maps of variables that are interconnected through a set of causal relations. Each relation has a numerical weight, representing intensity, direction, and sign (positive or negative) [17]. This computational model is then employed to simulate scenarios starting from differing conditions.

In accordance with Ozesmi and Ozesmi (2004), the approach can be divided into three macro-steps:

1. construction of the cognitive maps;
2. structural analysis of the cognitive maps;
3. simulation of possible future scenarios.

*Step 1.* In the present study, this step aims to identify: (i) the system variables and (ii) their causal relations and related weights. To carry out task (i), we conducted a desk analysis of the socio-economic context of southern Salento between February and March 2020 through an in-depth investigation of official and grey literature, including scientific publications, reports, journal articles, and websites. Through this search, we identified 19 variables. The preliminary list was then reviewed by three researchers (one agronomist with experience in *Xf* and two economists with expertise in the rural landscape), who identified the most relevant variables. Then, these variables were classified as: drivers (i.e., policy interventions) and effects (i.e., the environmental, social, and economic consequences of policy interventions). Task (ii) was performed using an ad hoc questionnaire. The questionnaire was administered to 13 representatives of local stakeholders (i.e., 7 local olive oil producers who had been affected by the epidemic, 1 representative of a local agricultural association, 4 researchers from Apulia universities with competencies in agricultural economics and agronomy, and 1 representative of the Agricultural Department of the Apulia Regional Government), who were asked to identify the existence of a causal relationship between each couple of variables, indicating its sign (positive when the increase in one variable causes an increase in the other; otherwise negative) and intensity (rated

on a three-point scale: 1 (*weak*), 2 (*medium*), 3 (*strong*)). Thus, stakeholders' perceptions were transformed into numerical data, which enabled us to create a cognitive map for each respondent in the form of an $n \times n$ adjacency matrix (where $n$ was the number of variables). Each $a_{i,j}$ matrix element represented the weight of the causal relation between variables $i$ and $j$. This information was then grouped into three kinds of maps, representing three levels of analysis: individual maps (13), category maps (4; i.e., Producers, Local Agricultural Association, Researchers, Regional Government), and an overall map that synthesised the perspectives of all stakeholder categories. The elements of the adjacency matrices were normalised in the range (−1, 1). As this process leads—in step 3—to the identification of the preferred policy mixes, this method lies in the more general revealed preferences approach [18].

*Step 2.* The structural analysis of the cognitive maps employed social network analysis (SNA) to calculate network and punctual indices [19]. Network indices capture salient aspects of the entire map, in terms of the number of connections, network density, and network centralization. Punctual indices characterize the relational profiles of variables, using out- and in-degree, as well as total degree indices. Table 1 describes these indices in more detail.

**Table 1.** Network and punctual indices employed to structurally analyze the cognitive maps.

| Index | Formula | Description |
|---|---|---|
| | Network indices | |
| Number of connections | Nc = \|L\| <br> L is the set of the map relations. | Number of relations between $n$ variables. |
| Network density | $D = \frac{Nc}{N(N-1)}$ <br> $N$ is the number of nodes. | Ratio between the number of actual connections and the maximum number of possible connections. This shows the map connectivity (i.e., how connected or sparse the map is). |
| Network centralization | $C = \frac{\sum_{i=1}^{N} Td_* - Td_i}{\sum_{i=1}^{N} Td_* - Td_i}$ <br> $Td_*$ is the degree of the most central node and $Td_i$ is the total degree as explained below. | Sum of differences between the degree of the most central node and the degrees of all other nodes, divided by the largest theoretical sum. This reflects the extent to which the network features one or more very central nodes. It ranges between 0 (i.e., completely democratic network, with influence evenly distributed across all nodes) and 1 (i.e., fully centralized network, with one variable influencing all others). This measure is calculated for both out- and in-degree indices. |
| | Punctual indices | |
| Out-degree | $od(v_i) = \sum_{k=1}^{N} \bar{a}_{ik}$ | Cumulative strength of connections ($a_{ik}$) exiting from variable $I$ and reaching the $k$ other variables. |
| In-degree | $id(v_i) = \sum_{k=1}^{N} \bar{a}_{ki}$ | Cumulative strength of connections ($a_{ki}$) entering variable $i$ and coming from other $k$ variables. |
| Total degree | $Td_i = od(v_i) + id(v_i)$ | Sum of the in- and out-degree indices. This shows how a variable is connected to others and the cumulative strength of its links. |

Variables were then classified into the following three types: *senders*, *transmitters*, and *receivers*. Senders have a positive out-degree and a zero in-degree index, and therefore, they serve as stimuli for the rest of the system (i.e., they send stimuli but do not receive any). Transmitters have both a positive out- and a positive in-degree index; they receive stimuli from senders and other transmitters, spreading them to the rest of the system. Receivers have a zero out-degree and a positive in-degree index, thus representing the ends of the system; they can be used to monitor the system's performance and the effects of any changes due to external pressure. In general, it is not necessary for all categories

of variables (i.e., senders, transmitters, receivers) to be present in real systems, as their distribution depends exclusively on stakeholder perceptions.

*Step 3.* In this step, we simulated possible future scenarios by means of FCMs. Simulations described the potential evolution of the system according to stakeholder perceptions, with the aim of verifying whether internal trends would converge toward system equilibrium.

To this end, data from step 1 were used to feed a computational model formed of an $n \times 1$ vector ($V$) (representing the initial values of the $n$ variables) and the $n \times n$ adjacency matrix ($A$) (containing the weights of the causal relations between variables). The model was run in an initialization phase ($t_0$) and a number of reiteration phases ($t_1, \ldots, t_n$). At $t_0$, the $n$ values of $V$ were arbitrarily set to 1 [12]. Subsequently, to determine the new state of $V$ at each $t$, $V$ was multiplied by $A^1$. This process was reiterated until $V$ stopped changing, thereby identifying the system as being in a *steady state* (s.s). This allowed us to predict how the system would evolve in the context of no external intervention. The calculation described above provided the basis for reasonable intervention scenarios, which simulated policy drivers—both individually and in combination. More specifically, simulations were created by manipulating the value of certain variables that could act as policy drivers. The selected variables were set to their maximum value (=1) to generate changes in the s.s. of other variables. Policy mixes were assessed according to the degree of change they effected in the s.s. of relevant variables. As stakeholder interests may differ from those of policy makers, we defined three assessment criteria:

(i)   *effectiveness*, operationalized as the sum of changes in the s.s. of the variables embodying policy objectives;

(ii)  *acceptability*, measured as the sum of changes in the s.s. of target variables (i.e., variables representing stakeholder viewpoints). As the policy perspective is unique, effectiveness was computed for the collective map, whereas complexive acceptability was calculated as the sum of each category's acceptability; and

(iii) *efficiency*, which concerned the number of policy drivers in the mix (assuming that mixes that achieve the same objective with fewer (costly) policy drivers are more efficient). Since no information was available for the costs of policy instruments, the underlying simplification assumed a comparable cost of implementation for all policy drivers. Section 3.3 reports the identification process of policy objectives and target variables.

Criteria (i), (ii), and (iii) informed the selection of the preferred policy mixes, as follows. First, measures of effectiveness and acceptability were combined into a synthetic measure of social utility, obtained as the algebraic sum of the effectiveness and acceptability values associated with each mix. This allowed us to identify mixes with the same level of utility (iso-utility sets). Second, the efficiency criterion was operationalized in consideration of three policy maker perspectives, based on the following principles:

- social utility maximization, which prefers mixes with the highest social utility and the fewest policy drivers;
- effectiveness maximization, which prefers mixes with the highest effectiveness and the fewest policy drivers; and
- constrained effectiveness maximization, which prefers mixes with the highest effectiveness, non-negative acceptability, and the fewest policy drivers.

## 3. Results

### 3.1. Construction of the Cognitive Maps

The final set of variables achieved from the desk analysis was composed of 16 variables, including 5 drivers and 11 policy effects (i.e., 3 environmental, 5 social, and 3 economic), as described in Table 2.

**Table 2.** Description of the system variables.

| N. | Variables | Abbreviation | Type | Description |
|---|---|---|---|---|
| 1 | Land-use planning | LUP | Policy driver | Aims at protecting soil and vegetation and mitigating hydrogeological risk. |
| 2 | Public participation | PUB | Policy driver | Promotes interaction between institutions and the community, enabling residents to contribute to decision making and planning. |
| 3 | Environmental regulation | ENR | Policy driver | Aims at protecting and preserving the environment by enforcing specific environmental norms for ecosystem conservation. |
| 4 | Income diversification | DIV | Policy driver | Includes measures to improve the diversification and stabilization of farmers' income. |
| 5 | Local development agencies | LOC | Policy driver | Refers to local action plans for rural development, promoted by local groups (i.e., public-private partnerships). |
| 6 | Monumental olive tree areas | MON | Effect (environmental) | Areas of high natural and ecological value characterized by the significant presence of centuries old olive trees with a trunk diameter of at least 1 metre and at least 1.5 metres of above-ground growth. |
| 7 | Ecosystem services | ECO | Effect (environmental) | Supply services (i.e., biomass produced by the ecosystem and consumed in the form of food, fibre, timber, etc.), regulatory services (which support ecosystem functioning by regulating the climate, pollutant uptake, water quality, etc.), support services (necessary for the provision of all other services, such as soil formation, photosynthesis, the nutrient cycle, etc.), and cultural services (i.e., intangible, spiritual, and intellectual benefits deriving from contact with nature). |
| 8 | Natural resources | NAT | Effect (environmental) | Natural resources, strictly speaking (i.e., those that are closely related to nature, including water, soil, flora-fauna, rivers, etc.) |
| 9 | Job opportunities | JOB | Effect (social) | Ability of the local labour system to provide suitable jobs for local communities, including marginalized people |
| 10 | Place branding | BRA | Effect (social) | Ability of a territory to develop a competitive identity according to its authentic characteristics and vocation. |
| 11 | Social and cultural inertia | SCI | Effect (social) | Community resistance to change (i.e., to adopt adaptation and mitigation measures directed at landscape regeneration), based on individual and group social habits. |
| 12 | Openness | OPE | Effect (social) | Transparent local decision making through the honest and effective disclosure of relevant information (i.e., how governments conduct public business and allocate resources) to the local community. |
| 13 | Environmental awareness | ENV | Effect (social) | Increased capacity of the local community to understand the fragility of their environment and the importance of its protection. |
| 14 | Production loss | PRO | Effect (economic) | Reduction in olive oil production due to the *Xf* epidemic. |
| 15 | Tourism | TUR | Effect (economic) | Whole set of multipurpose activities and services to sustain tourist flows to the relevant area. |
| 16 | Agricultural sector loss of competitiveness | COM | Effect (economic) | Loss of comparative and competitive advantage of the agriculture sector due to higher costs, reduced resources, and reduced production quality, caused by the Xf epidemic. |

The questionnaire described above (second task of step 1) was administered between April and May 2020. Respondents were first contacted via telephone and informed about the research aims. Subsequently, those who agreed to be interviewed were asked to fill in, according to their views, the 16 × 16 matrix of variables and the weights of possible causal relationships. Respondents received this material by email and were provided with telephone support to perform the task. Figure 2 reports the category and overall cognitive maps.

Producers                                    Local Agricultural Association

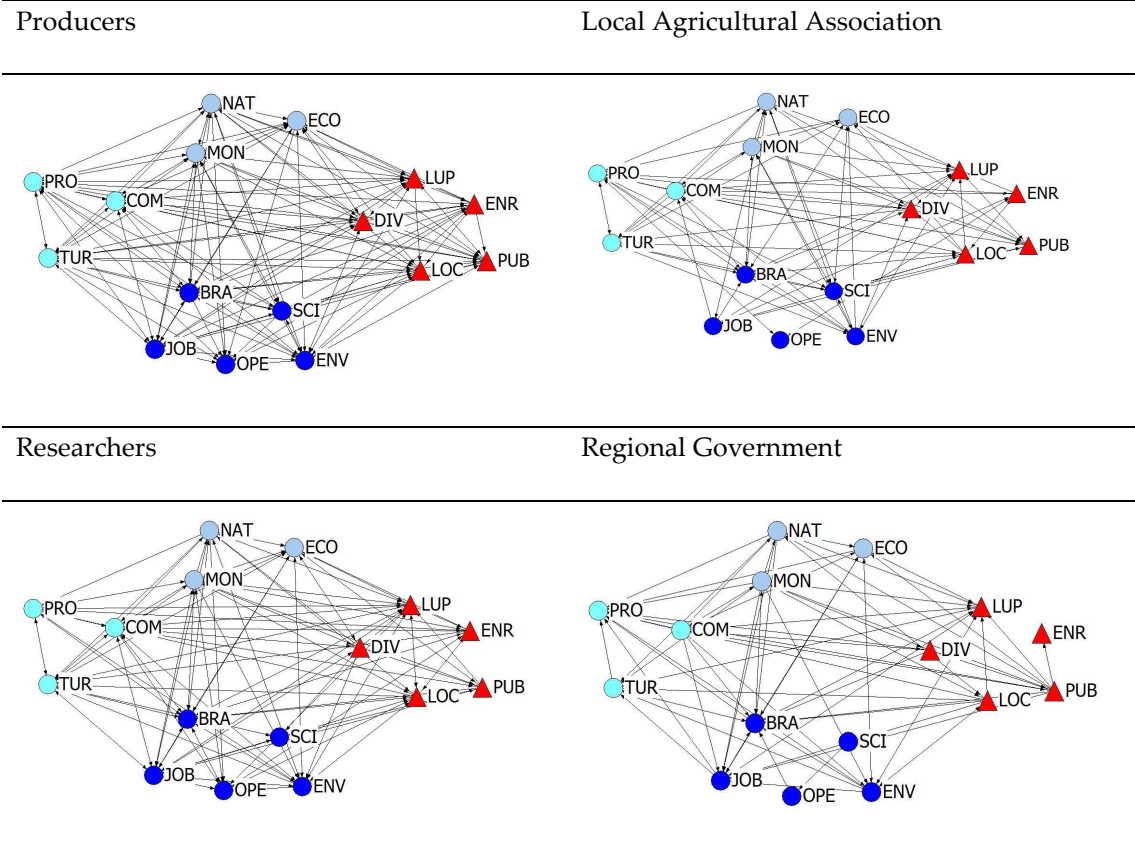

Researchers                                  Regional Government

Overall map*

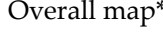

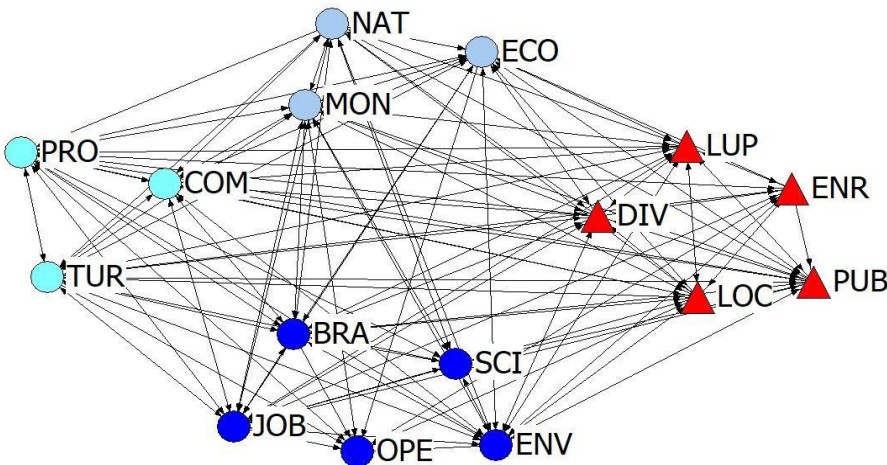

\* The overall map was obtained by combining the four category maps. Legend: red triangles = policy drivers; circles = effects (powder blue = environmental; blue = social; light blue = economic).

**Figure 2.** Category and overall cognitive maps.

*3.2. Structural Analysis of the Cognitive Maps*

Table 3 reports the network indices.

**Table 3.** Network indices.

| Maps | Number of Connections | Density | Out-Degree Centralization (OdC) | In-Degree Centralization (IdC) |
|---|---|---|---|---|
| Producers | 229 | 0.95 | 0.05 | 0.05 |
| Local Agricultural Association | 145 | 0.6 | 0.35 | 0.28 |
| Researchers | 179 | 0.75 | 0.27 | 0.27 |
| Regional Government | 97 | 0.4 | 0.42 | 0.49 |
| Overall | 229 | 0.95 | 0.05 | 0.05 |

The network indices reveal that stakeholders perceived the system as very interconnected, with only one category with fewer than 100 connections. This was reflected in an overall map density of 0.95.

The *Local Agricultural Association* and the *Regional Government* maps produced the highest centralization measures. On the one hand, this reflects the significant presence of control forces in the system; however, in the case of the *Regional Government* map, the high OdC and even higher IdC suggest that most of the stimuli flowed toward one, or only a few variables, which could be considered as the end of the system, and therefore, markers of the policy drivers' action. In contrast, the *Local Agricultural Association* map exhibited a lower IdC than OdC, indicating that the stimuli sent by the controlling forces reach all parts of the system fairly equally. The *Researchers* map shows a perfect balance of OdC and IdC. Finally, the *Producers* network was almost complete and very decentralized, reflecting a very high density; as this was the largest map, it also represented the lower bound of the overall map, which exhibited almost the same metrics.

The punctual indicators (Table A1) formed a very intricate system of (almost exclusively) transmitters, representing the "connective fabric" of the structure [6]. No senders existed, whereas only one receiver (JOB) emerged in the *Regional Government* map. This means that each element was able to both receive and spread stimuli to other parts of the system. Moreover, the analysis revealed that some variables (i.e., those with higher centrality, marked in green) played the role of transmitter more markedly than others.

To further explore the role played by each variable in collecting and sending input, differences between the in- and out-degree indices were calculated, following relativization to make the maps comparable. Table A2 reports these differences in the form of a visual representation, with blue (red) bars indicating whether the out-degree index was higher (lower) than the in-degree index, and bar size expressing the width of the difference. Accordingly, transmitters were divided into two groups: net stimuli spreaders (blue bars) and net stimuli recipients (red bars). Looking at the *overall map* (column 5), as expected, policy drivers were largely stimuli spreaders, since they sent much more input than they received. This was particularly the case for LOC and DIV, which were perceived as net spreaders by all stakeholders, excluding producers. A similar (albeit more limited) influence was exerted by LUP and ENR, which were mainly perceived as spreaders. PUB represented an exception, since it was considered more subject to receiving external stimuli than sending stimuli, being the only driver with a red bar.

In contrast to the policy drivers, policy effects were mainly considered net stimuli recipients. For instance, JOB was perceived as a net stimuli recipient by all stakeholder categories (columns 1–4), and wide agreement among actors was also registered for NAT, BRA, and COM. The opposite dynamics emerged for MON and ENV, which were mainly perceived as net stimuli spreaders.

### 3.3. Simulation of Possible Future Scenarios

To assess the effectiveness and acceptability of policy mixes, we first identified policy objectives and target variables.

System variables related to landscape regeneration (i.e., MON, ECO, and NAT) were thought to represent environmental interests and, accordingly, selected as policy objectives.

On the other side, variables thought to represent the interests of each stakeholder category were as follows:

- PRO and COM for the *Producers* map, which aimed at reducing the negative impact of the *Xf* epidemic on production, and indirectly, on the competitiveness of the agricultural sector;
- PRO and COM for the *Local Agricultural Association* map, as they are representative of all actors involved in the sector;
- ECO and COM for the *Researchers* map, as the experts interviewed were specialised in agronomy (with a focus on ecosystem services) and agricultural economics; and
- NAT and COM for the *Regional Government* map, in line with the most relevant functions of the Agriculture Directorate of the Regional Government (i.e., the sustainable management and protection of forests and natural resources and the competitiveness of agri-food chains).

We then focused on the effects of the policy drivers on the aforementioned variables, in terms of change in s.s. To this end, we simulated all possible solutions (i.e., all 31 combinations of policy drivers): five referring to each single policy driver, ten referring to two-driver policy mixes, ten referring to three-driver policy mixes, five referring to four-driver policy mixes, and a final policy mix formed of all policy drivers. Figure 3 shows policy mixes according to their effectiveness, acceptability, and efficiency.

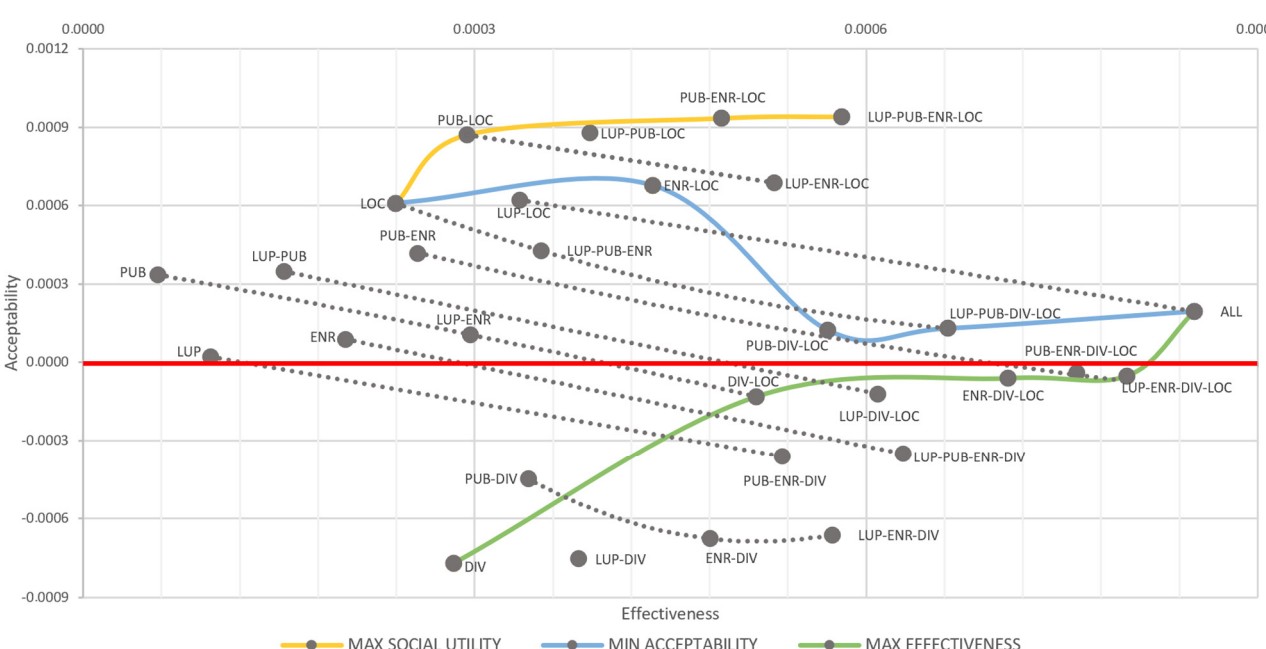

**Figure 3.** Effective, acceptable, and efficient policy mixes.

In the figure, points are associated with specific levels of effectiveness (horizontal axis) and acceptability (vertical axis). Effectiveness increases along the X axis (moving rightwards), whereas acceptability increases along the Y axis (moving upwards). Although all mixes exhibit positive effectiveness, some have negative acceptability (situated below the red line). Mixes with the same level of social utility are connected via a dotted line, forming iso-utility sets (with utility increasing the further rightward and upward they move). Colored curves identify the most efficient mixes (i.e., those achieving the objectives with the fewest policy drivers) according to, respectively, the principles of:

- social utility maximization (yellow curve, connecting mixes with the most straightforward *upward and rightward* trajectory and the fewest policy drivers);
- effectiveness maximization (green curve, connecting mixes with the most straightforward *rightward* trajectory and the fewest policy drivers); and

- constrained effectiveness maximization, subject to a minimum (non-negative) acceptability (light blue curve, connecting mixes *above* the red line with the most straightforward *rightward* trajectory and the fewest policy drivers).

As is evident in Figure 3, the set of efficient mixes change according to the policy maker's perspective. Exceptions are represented by LOC (a potential efficient driver in the case of both social utility maximization and constrained effectiveness maximization) and ALL (a potentially efficient mix in the case of both constrained effectiveness maximization and effectiveness maximization).

Since the yellow curve (i.e., social utility maximization) reflects both effectiveness and acceptability criteria, it resembles a typical frontier, moving constantly upward and rightward. A trade-off between acceptability and effectiveness may occur if policy makers follow either of the other two maximization principles. Indeed, the light blue curve shows that the increase in effectiveness associated with movement from the two- to three-driver mix (i.e., from ENR-LOC to PUB-DIV-LOC) corresponds to a large acceptability reduction. What is the cause of this trade-off? It can be traced back to DIV, which is the most effective but least acceptable individual policy driver. According to the Regional Government, the effectiveness of DIV stems from its potential to trigger the multifunctionality of agriculture through complementary activities such as agri-tourism, social agriculture, educational farms, local services provided, and typical crafts. This generates greater environmental conservation by farmers, with a positive impact on landscape regeneration. In contrast, Producers and Local Agricultural Associations view DIV as unacceptable, mainly due to concerns that this driver could lead to too much change in producer practices.

Finally, considering the two-driver mix (DIV-LOC), the green curve is superior to the blue curve, as it involves greater efficiency in the landscape regeneration process with a minimally negative level of acceptability. A similar result is evident for the three-driver mix on the green curve (ENR-DIV-LOC), which achieves higher effectiveness than even the four-driver mix on the light-blue curve (LUP-PUB-DIV-LOC), in the face of a minimum gain in acceptability.

## 4. Conclusions

In this work, FCMs were applied to identifying sustainable policy actions for renewing the rural landscape of Salento, which has been significantly affected by the Xf epidemic, by including the criteria of effectiveness and acceptability in the analysis. To this end, actors who are representative of local stakes were interviewed in order to obtain a clear perception of the territory, along with a technical and social viewpoint. This system representation was used to build a scenario analysis based on different policy actions.

Such a methodology allows policy makers to easily identify the most effective and acceptable protection/regeneration policy mixes and to visualise the perception gap among the various stakeholders' groups. Moreover, it is useful to find out which policy mixes match more than one criterion. When this does not occur, a trade-off between effectiveness and acceptability should be considered. Indeed, the adoption of the minimum acceptability perspective (i.e., the effectiveness maximization constrained by non-negative acceptability) represents a way for this trade-off to emerge, providing a rational basis for negotiation and reducing, at the same time, the risk of policy failure.

Although the analysis lacked information on budgetary constraints and the preferred maximization principle, such information is already available to policy makers, who are in charge of allocating financial resources according to a specific political orientation. In other words, following the presented framework, the public decision maker need only choose: (i) the curve that best represents his or her political orientation and (ii) the mix that reflects the highest utility, given the available budget.

Due to the unavailability of information on the cost of policy instruments, the paper lacks a policy cost analysis; however, this limitation does not negatively impact the robustness of the investigation or the methodological framework, since policy mixes can be

easily reordered according to cost. Further empirical analyses might include an appraisal of policy costs, which is beyond the scope of this paper.

**Author Contributions:** Conceptualization, A.L. and E.S.; methodology, A.L. and E.S.; software, A.L.; validation, A.L. and E.S.; formal analysis, A.L. and E.S.; investigation, A.L. and E.S.; resources, A.L.; data curation, A.L.; writing—original draft preparation, A.L. and E.S.; writing—review and editing, A.L. and E.S.; visualization, A.L. and E.S.; supervision, A.L.; project administration, A.L.; funding acquisition, A.L. All authors have read and agreed to the published version of the manuscript.

**Funding:** This research and the APC were funded by the Annual Research Project of University of Foggia (Italy) (2019) "Participatory scenario building for landscape regeneration in rural areas affected by the Xylella epidemic" (Rector decree no. 619/2020).

**Conflicts of Interest:** The authors have no conflict of interest to declare. All co-authors have seen and agree with the contents of the manuscript and there is no financial interest to report. We certify that the submission is original work and is not under review at any other publication.

## Appendix A

**Table A1.** Punctual indices.

| N. | Concepts | Producers | | | Local Agric. Assoc. | | | Researchers | | | Apulia Region Gov. | | | Overall | | |
|---|---|---|---|---|---|---|---|---|---|---|---|---|---|---|---|---|
| | | Outd. | Ind. | Central. | Outd. | Ind. | Central. | Outd. | Ind. | Central. | Outd. | Ind. | Central. | Outd. | Ind. | Central. |
| 1 | Land-use planning | 7.82 | 6.18 | 14.00 | 3.33 | 8.00 | 11.33 | 5.50 | 4.90 | 10.40 | 4.33 | 5.33 | 9.67 | 7.39 | 6.57 | 13.96 |
| 2 | Public participation | 2.53 | 4.94 | 7.47 | 2.33 | 6.33 | 8.67 | 2.50 | 2.50 | 5.00 | 6.67 | 2.00 | 8.67 | 2.96 | 4.5 | 7.46 |
| 3 | Environmental regulation | 6.94 | 5.47 | 12.41 | 3.67 | 4.67 | 8.33 | 3.30 | 2.70 | 6.00 | 3.33 | 3.33 | 6.67 | 5.79 | 4.79 | 10.57 |
| 4 | Income diversification | 2.82 | 3.12 | 5.94 | 9.67 | 4.67 | 14.33 | 2.90 | 2.70 | 5.60 | 3.67 | 2.67 | 6.33 | 3.46 | 3.18 | 6.64 |
| 5 | Local development Agencies | 4.88 | 5.35 | 10.24 | 10.00 | 4.00 | 14.00 | 5.30 | 3.30 | 8.60 | 6.00 | 1.00 | 7.00 | 5.57 | 4.32 | 9.89 |
| 6 | Monumental olive trees areas | 5.71 | 4.18 | 9.88 | 5.67 | 3.00 | 8.67 | 5.00 | 4.70 | 9.70 | 2.00 | 4.67 | 6.67 | 5.5 | 4.39 | 9.89 |
| 7 | Ecosystem services | 5.71 | 5.65 | 11.35 | 8.00 | 7.67 | 15.67 | 3.20 | 5.20 | 8.40 | 3.33 | 4.67 | 8.00 | 5.54 | 5.82 | 11.36 |
| 8 | Natural resources | 5.35 | 6.18 | 11.53 | 9.00 | 5.67 | 14.67 | 3.80 | 4.30 | 8.10 | 3.33 | 4.00 | 7.33 | 5.5 | 5.68 | 11.18 |
| 9 | Job opportunities | 4.47 | 4.82 | 9.29 | 6.33 | 9.67 | 16.00 | 1.70 | 2.90 | 4.60 | 0.00 | 7.33 | 7.33 | 3.07 | 4.86 | 7.93 |
| 10 | Place branding | 6.06 | 5.53 | 11.59 | 8.00 | 9.00 | 17.00 | 2.20 | 3.40 | 5.60 | 3.33 | 6.33 | 9.67 | 5.39 | 5.43 | 10.82 |
| 11 | Social and cultural inertia | 2.12 | 3.29 | 5.41 | 8.67 | 8.67 | 17.33 | 4.00 | 2.70 | 6.70 | 5.33 | 2.00 | 7.33 | 1.64 | 2.96 | 4.61 |
| 12 | Openness | 4.06 | 3.71 | 7.76 | 4.33 | 5.67 | 10.00 | 3.30 | 1.60 | 4.90 | 1.33 | 2.67 | 4.00 | 4.11 | 3.64 | 7.75 |
| 13 | Environmental awareness | 5.65 | 5.00 | 10.65 | 7.00 | 9.00 | 16.00 | 4.80 | 4.80 | 9.60 | 7.00 | 1.00 | 8.00 | 6.43 | 5.46 | 11.89 |
| 14 | Production loss | 3.35 | 3.12 | 6.47 | 9.33 | 7.33 | 16.67 | 1.80 | 1.90 | 3.70 | 0.33 | 1.00 | 1.33 | 2.00 | 2.39 | 4.39 |
| 15 | Tourism | 6.12 | 5.71 | 11.82 | 5.00 | 5.00 | 10.00 | 2.70 | 3.70 | 6.40 | 1.67 | 4.00 | 5.67 | 4.86 | 5.5 | 10.36 |
| 16 | Agricultural sector loss of competitiveness | 1.53 | 2.88 | 4.41 | 3.00 | 5.00 | 8.00 | 1.90 | 2.60 | 4.50 | 1.33 | 1.00 | 2.33 | 1.93 | 1.64 | 3.57 |

Table A2. Relativized in- and out-degrees.

| N. | Variables Name | Type | (1) Producers Rel.Outd. | Rel.Ind. | Diff. | (2) Local Agric. Assoc. Rel.Outd. | Rel.Ind. | Diff. | (3) Researchers Rel.Outd. | Rel.Ind. | Diff. | (4) Apulia Region Gov. Rel.Outd. | Rel.Ind. | Diff. | (5) Overall Rel.Outd. | Rel.Ind. | Diff. |
|---|---|---|---|---|---|---|---|---|---|---|---|---|---|---|---|---|---|
| 1 | Land-use planning | Policy Drivers | 1.00 | 0.79 | 0.21 | 0.33 | 0.80 | −0.47 | 1.00 | 0.89 | 0.11 | 0.59 | 0.73 | −0.14 | 1.00 | 0.89 | 0.11 |
| 2 | Public participation | Policy Drivers | 0.32 | 0.63 | −0.31 | 0.23 | 0.63 | −0.40 | 0.45 | 0.45 | 0.00 | 0.91 | 0.27 | 0.64 | 0.40 | 0.61 | −0.21 |
| 3 | Environmental regulation | Policy Drivers | 0.89 | 0.70 | 0.19 | 0.37 | 0.47 | −0.10 | 0.60 | 0.49 | 0.11 | 0.45 | 0.45 | 0.00 | 0.78 | 0.65 | 0.14 |
| 4 | Income diversification | Policy Drivers | 0.36 | 0.40 | −0.04 | 0.97 | 0.47 | 0.50 | 0.53 | 0.49 | 0.04 | 0.50 | 0.36 | 0.14 | 0.47 | 0.43 | 0.04 |
| 5 | Local development Agencies | Policy Drivers | 0.62 | 0.68 | −0.06 | 1.00 | 0.40 | 0.60 | 0.96 | 0.60 | 0.36 | 0.82 | 0.14 | 0.68 | 0.75 | 0.58 | 0.17 |
| 6 | Monumental olive trees areas | Impacts (env.) | 0.73 | 0.53 | 0.20 | 0.57 | 0.30 | 0.27 | 0.91 | 0.85 | 0.05 | 0.27 | 0.64 | −0.36 | 0.74 | 0.59 | 0.15 |
| 7 | Ecosystem services | Impacts (env.) | 0.73 | 0.72 | 0.01 | 0.80 | 0.77 | 0.03 | 0.58 | 0.95 | −0.36 | 0.45 | 0.64 | −0.18 | 0.75 | 0.79 | −0.04 |
| 8 | Natural resources | Impacts (env.) | 0.68 | 0.79 | −0.11 | 0.90 | 0.57 | 0.33 | 0.69 | 0.78 | −0.09 | 0.45 | 0.55 | −0.09 | 0.74 | 0.77 | −0.02 |
| 9 | Job opportunities | Impacts (social) | 0.57 | 0.62 | −0.04 | 0.63 | 0.97 | −0.33 | 0.31 | 0.53 | −0.22 | 0.00 | 1.00 | −1.00 | 0.42 | 0.66 | −0.24 |
| 10 | Place branding | Impacts (social) | 0.77 | 0.71 | 0.07 | 0.80 | 0.90 | −0.10 | 0.40 | 0.62 | −0.22 | 0.45 | 0.86 | −0.41 | 0.73 | 0.73 | −0.01 |
| 11 | Social and cultural inertia | Impacts (social) | 0.27 | 0.42 | −0.15 | 0.87 | 0.87 | 0.00 | 0.73 | 0.49 | 0.24 | 0.73 | 0.27 | 0.45 | 0.22 | 0.40 | −0.18 |
| 12 | Openness | Impacts (social) | 0.52 | 0.47 | 0.04 | 0.43 | 0.57 | −0.13 | 0.60 | 0.29 | 0.31 | 0.18 | 0.36 | −0.18 | 0.56 | 0.49 | 0.06 |
| 13 | Environmental awareness | Impacts (social) | 0.72 | 0.64 | 0.08 | 0.70 | 0.90 | −0.20 | 0.87 | 0.87 | 0.00 | 0.95 | 0.14 | 0.82 | 0.87 | 0.74 | 0.13 |
| 14 | Production loss | Impacts (econ.) | 0.43 | 0.40 | 0.03 | 0.93 | 0.73 | 0.20 | 0.33 | 0.35 | −0.02 | 0.05 | 0.14 | −0.09 | 0.27 | 0.32 | −0.05 |
| 15 | Tourism | Impacts (econ.) | 0.78 | 0.73 | 0.05 | 0.50 | 0.50 | 0.00 | 0.49 | 0.67 | −0.18 | 0.23 | 0.55 | −0.32 | 0.66 | 0.74 | −0.09 |
| 16 | Agricultural sector loss of competitiveness | Impacts (econ.) | 0.20 | 0.37 | −0.17 | 0.30 | 0.50 | −0.20 | 0.35 | 0.47 | −0.13 | 0.18 | 0.14 | 0.05 | 0.26 | 0.22 | 0.04 |

## Note

<sup>1</sup> After multiplying, values in *V* underwent a logistic transformation to keep their values within the range [−1,1]. For further details on this methodology, see Lopolito et al. (2020).

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
