# Peer review of "Designing Policy Mixes to Address the World’s Worst Devastation of a Rural Landscape Caused by Xylella Epidemic"

_land, doi:10.3390/land11050763_

Round 1

Reviewer 1 Report

Dear authors,

I found your article title and summary very interesting and therefore I decided to become your reviewer. Meanwhile, while reding it I realised that you have a very agricultural and economic approach. If you are dealing with rural landscape revitalisation and moreover working with cognitive maps, visual identity of the site must be very important. Therefore, you need to upgrade at least your introduction and conclusion (I would like that you do in other parts of article as well) with visual characterisation of the investigated site. Visual identification is crucial for some aspects of your exploration because its is directly connected to tourism, property value and quality of life.

Kind regards

Author Response

Many thanks for your valuable comments. We agree about the relevance of visual characterisation for improving the manuscript. The manuscript already includes figure 1 to show Salento’s landscape devastation due to Xylella fastidiosa infection. Due to copyright reasons, we cannot add any other picture useful to characterize the landscape and its degradation.

Reviewer 2 Report

- the summary is complete
- the introduction is very well structured, with satisfactory references
- the methodology is well described 
- the results and conclusions are well presented

- lines 206-207 must be removed

  • both rows must be removed

Author Response

Many thanks for your revision, we addressed your point by removing both rows 206-207.

Reviewer 3 Report

The work presented is original, convenient and methodologically solidly constructed.
In addition, it is considered as an applied work that can be used for decision-making by public actors.
Two aspects should be improved.
First, a more and better discussion of the results presented. In the first paragraphs of this section, issues that would be part of the materials and methodology section are addressed.
Second, an improvement of the conclusions since those that are included are not. They are reflections on the limitations of the work carried out and future lines of research, not conclusions.

Author Response

Reply: We wish to thank you for your suggestions.

We revised the results section by moving the first paragraph to materials and methods. A more in-depth discussion of the results has been provided by widening the conclusions.